# Valorization of Papaya By-Products: Bioactive Potential of Peel and Seeds and Their In Vitro Bioavailability

**DOI:** 10.3390/foods14223885

**Published:** 2025-11-13

**Authors:** Sayonara Reyna, María de Guía Córdoba, María Ángeles Rivas, Iris Gudiño, María Vázquez-Hernández, Víctor Otero-Tuárez, Rocío Casquete

**Affiliations:** 1Nutrición y Bromatología, Escuela de Ingenierías Agrarias, Universidad de Extremadura, Avd. Adolfo Suárezs/n, 06007 Badajoz, Spain; sayonara.reyna@gmail.com (S.R.); mrivasm@unex.es (M.Á.R.); igudino@unex.es (I.G.); mariavh@unex.es (M.V.-H.); rociocp@unex.es (R.C.); 2Instituto Universitario de Investigación en Recursos Agrarios (INURA), Universidad de Extremadura, Avd. de la Investigación, 06006 Badajoz, Spain; 3Facultad de Ciencias de la Vida y Tecnología, Universidad Laica Eloy Alfaro de Manabí, Ciudadela Universitaria, Manta 130802, Ecuador; victor.otero@uleam.edu.ec

**Keywords:** papaya by-products, phenolic compounds, antioxidant activity, antimicrobial activity, in vitro digestion

## Abstract

Papaya (*Carica papaya* L.) processing generates by-products that can serve as sustainable sources of bioactive compounds. This study aimed to extract and characterize the bioactive compounds from the peel and seeds of different papaya varieties and evaluate their antioxidant and antimicrobial potential, as well as their behavior under simulated digestion. The results indicated that Maradol seeds possessed the highest total phenolic content and antioxidant values, demonstrating superior compositional and functional profiles, and that seed extracts overall had greater antibacterial efficacy than peel extracts, with Hawaiian seed extracts exhibiting the greatest overall inhibition. Furthermore, under simulated gastrointestinal conditions, the combined extracts from peel and seeds effectively preserved phenolics through the gastric and intestinal phases and notably enhanced the generation of acetate and propionate during colonic fermentation. These findings robustly substantiate the functional valorization of papaya by-products and suggest that selecting extracts based on their specific bioactive profiles can significantly enhance their applications as natural, functional ingredients in the food industry.

## 1. Introduction

Papaya (*Carica papaya* L.) is a tropical fruit of great nutritional, economic, and industrial relevance, whose main use is centered on the consumption of its edible pulp. Globally, its production remains relatively stable, with estimated values ranging between 14 and 15 million tons per year, with India as the leading producer, followed by the Dominican Republic, Mexico, and Brazil [1]. In Ecuador, papaya cultivation is primarily concentrated in the coastal region, with the largest production areas located in the provinces of Esmeraldas, Santo Domingo de los Tsáchilas, Los Ríos, Guayas, Santa Elena, Manabí, and El Oro, which supply both the domestic market and export channels [2]. However, a considerable fraction of the fruit, mainly the peel and seeds, which account for approximately 20% to 25% of the total fruit weight, is often discarded during processing, leading to environmental impacts and a significant loss of valuable resources with high potential for valorization [3]. In the last decade, these by-products have garnered attention due to their high dietary fiber content [4] and the wide range of bioactive compounds they contain, such as phenols and flavonoids [5], carotenoids [6] and enzymes [7]. This has opened opportunities for their incorporation as functional ingredients in food, cosmetics, and pharmaceutical products [8].

Papaya peel and seed composition vary with cultivar, ripening stage, and extraction method [9,10,11]. These differences are evident in both the phenolic profiles and concentrations; in the Formosa and Aliança variety comparison, the hulls show significantly higher total phenolics and flavonoids in the Aliança variety, while the seeds show comparable total phenolics but significantly different flavonoids [12]. Likewise, analyses considering two pulp colors (yellow vs. red) across ripening stages reported distinct profiles for pulp, peel, and seeds [5]. Regarding processing, optimization of extraction conditions in seeds increased polyphenol recovery and antioxidant activity [13], while microwave-based approaches have been optimized in the shell to maximize phenolics and antioxidant capacity [14].

Recent studies highlight the dual functional potential of papaya by-products—especially peel and seeds—as natural ingredients to enhance food quality and safety through antioxidant and antimicrobial effects. Papaya peel and seed extracts are notable antioxidants for food preservation: in a meat model, peel extract reduced lipid oxidation and maintained quality metrics during storage [15]. Comparative studies also report stronger antioxidant activity in peel than in seed across solvent fractions, reinforcing peel’s role as a natural antioxidant source [16]. Papaya by-product extracts exhibit broad-spectrum antibacterial effects against foodborne bacteria (*Escherichia coli*, *Staphylococcus aureus*, *Bacillus subtilis*, among others) [17,18]. One of the key compounds involved in the antimicrobial activity was benzyl isothiocyanate (BITC), derived from papaya seeds [19], which showed potent antibacterial and anti-biofilm effects and, notably, inhibited bacterial growth in a beef model [20].

Beyond their content and in vitro functionality, the physiological relevance of bioactive compounds depends on their stability, release, and bioaccessibility and bioavailability during digestion. In papaya matrices, the bioaccessibility of carotenoids from by-products improves with oil-in-water emulsions [21] and with processing treatments that increase the stability and bioaccessibility of antioxidants in papaya purée [22]. Additionally, colonic fermentation of these matrices can modulate gut microbial communities and promote the production of short-chain fatty acids (SCFAs), which are considered key mediators of downstream physiological effects [23].

In this context, the present work aims to extract and characterize the bioactive compounds from papaya peel and seeds across different varieties and evaluate their antioxidant and antimicrobial potential as well as their behaviour under simulated digestion, tracking changes in antioxidant activity and phenolic content, and considering possible interactions with the microbiota, including SCFA production. Although papaya phenolic components have been studied, an integrated and variety-specific assessment is still lacking due to pronounced intervarietal and tissue-dependent heterogeneity; here, we address this gap by characterizing peel and seed matrices and coupling compositional analyses with antioxidant/antimicrobial assays and simulated gastrointestinal digestion-colonic fermentation to quantify bioaccessibility and SCFA outcomes.

## 2. Materials and Methods

### 2.1. Plant Material

Papaya fruits (varieties Tainung, Maradol, Solo, and Hawaiian) were sourced as follows: Tainung, Maradol, and Solo were collected in Paján, Manabí, Ecuador (GPS, WGS84: 1°34′ S, 80°25′ W; −1.57, −80.42), and the Hawaiian variety was obtained in Manta, Manabí, Ecuador (GPS, WGS84: 0°57′00″ S, 80°42′58″ W; −0.95, −80.716). A total of 12 fruits per variety were collected at a semi-ripe stage (N = 12 per variety). Fruits were transported to the laboratory at Universidad Laica Eloy Alfaro de Manabí (Manta, Ecuador) at ambient temperature (25 ± 2 °C) under natural ventilation, protected from direct sunlight and heat sources, and held under the same conditions until processing.

### 2.2. Bacterial Strains

For this study, six strains from the Spanish Type Culture Collection (*Staphylococcus aureus* CECT 976, *Bacillus cereus* CECT 131, *Listeria monocytogenes* CECT 911, *Listeria innocua* CECT 910, and *Salmonella choleraesuis* CECT 4395) were used.

### 2.3. Extraction and Determination of Total Phenolic Compounds Content

Ultrasound-assisted extraction (UAE) was performed using 80% ethanol (*v*/*v*) as the solvent. The extraction was applied to peel and seed by-products from each papaya variety (Tainung, Maradol, Solo, and Hawaiian) [24]. To do this, 60 mL of the extraction solution was combined with 10 g of dry sample, and the mixture was then immersed in an ultrasonic bath (40 kHz) at 45 °C for an hour. After filtering the extract via filter paper, the previously mentioned procedure was repeated with an additional 60 mL of the extraction solution added to the remaining solid. An aqueous extract was produced by removing the ethanol from the extract by evaporating it after the second filtration. Ultimately, what was extracted was kept at −20 °C for further analysis.

The Folin–Ciocalteu colorimetric method was employed to quantify total phenolic compounds, as described by Casquete et al. [24]. Values were calculated from a gallic acid calibration curve and reported as gallic acid equivalents (GAEs). For each sample, a 25 mL flask was prepared. The following were added sequentially: 10 mL MilliQ water, 0.5 mL of sample, 1 mL Folin–Ciocalteu reagent, and 2 mL saturated sodium carbonate (Na_2_CO_3_) solution. The final volume was adjusted to 25 mL with MilliQ water, and thorough mixing was ensured. Samples were then incubated at ambient temperature in darkness for 60 min before their absorbance was recorded at 760 nm. The concentration of total phenolics in each extract was calculated from the gallic acid calibration curve, with results reported in mg/L.

### 2.4. Identification of Bioactive Compounds

To identify the compounds within each extract, samples were first prepared by diluting them in LC-MS grade methanol to a final concentration of 100 ppm. Following filtration through a 0.45 μm syringe filter, these prepared samples were injected into an HPLC system coupled to a mass spectrometer (HPLC-QTOF Agilent Model G6530, Agilent Technologies, Palo Alto, CA, USA). The separation was achieved using a C_18_ column (4.6 × 150 mm, ×4.8 μm). Detection and initial identification of the bioactive compounds were performed with a quadrupole time-of-flight (Q-TOF) tandem mass analyzer, utilizing an electrospray ionization (ESI) source in negative mode. Provisional compound identities were assigned through consultation with the MassBank database with Qualitative Analysis, version B.07.00 [25,26].

### 2.5. Antioxidant Determination of the Phenolic Extracts

The antioxidant capacity of the samples was evaluated by two methods were utilized: the 2,2-diphenyl-1-picrylhydrazyl (DPPH) assay and the 2,2′-azinobis (3-ethylbenzothiazoline-6-sulfonic acid) (ABTS) radical scavenging assay. The DPPH method followed Kondo et al. [27]: 2950 μL of DPPH reagent was combined with 50 μL of sample in a cuvette, reacted for 30 min at room temperature in the dark, and then measured spectrophotometrically at 515 nm. For the ABTS method, based on Cano and Arnao [28] with minor adjustments, 1 mL of ABTS radical cation was mixed with 20 μL of the sample in a cuvette. Initial absorbance was recorded at 730 nm and compared against the absorbance after a 20 min reaction. Trolox served as the standard, with results expressed as mg of Trolox per 100 g of extract.

### 2.6. Antimicrobial Activity Determination

The antibacterial activity of the obtained extracts was assessed by examining their influence on microbial growth within a liquid culture system. Brain Heart Infusion (BHI) broth was prepared following manufacturer guidelines and used for bacterial cultivation, with initial growth occurring at 37 °C for 24 h. For the assay, each microorganism was adjusted to 10^5^ CFU/mL (2% *v*/*v* suspension) and inoculated into individual wells of a microtiter plate. Each well also received varying concentrations of the different extracts. Once the microtiter plate was filled with samples and both negative and positive controls (the inoculum of each microorganism cultured under the same conditions but in the absence of any extract), the microtiter plate was transferred to a plate reader and incubated at 37 °C for 24 h. Throughout this incubation, absorbance at 560 nm was automatically measured hourly, providing real-time data on microbial proliferation or its inhibition. The findings are presented as the percentage inhibition of the extract against each tested microorganism. The percentage of growth inhibition was then calculated relative to this control using the formula: [(A control − A sample)/A control] × 100, where A control is the absorbance of the control and A sample is the absorbance of the well containing the extract.

### 2.7. Simulated Gastrointestinal Digestion

Simulated gastrointestinal digestion was performed using a three-stage BIOSTAT^®^ A bioreactor system (Sartorius, Germany), following the protocol described by Rivas et al. [29], adapted for this study. The reactors (37 °C, 150 rpm, <0.5% O_2_), connected in series by peristaltic pumps, mimicked physiological pH conditions: 2.5 (stomach), 6.5 (small intestine), and a gradient of 5.7 to 6.8 (colon). The samples consisted of 250 mL of papaya polyphenol extract (500 ppm in 1% gelatin/PBS), prepared as a 50:50 (*v*/*v*) mixture of peel and seed extracts of the Hawaiian variety, selected for its good results based on its functional properties; controls contained only PBS. The colon reactor was inoculated with 25 mL of Man, Rogosa, and Sharpe (MRS) broth and 10 mL of fecal slurry, followed by a 12 h stabilization period. Digestive fluids were added automatically, including pepsin (4.5% in the gastric phase), pancreatin (3%), and bile salts (7.5%) in the intestinal phases (all reagents from Sigma-Aldrich, St. Louis, MO, USA). Each sample type was processed in duplicate. Aliquots were collected in triplicate at baseline, post-gastric, post-small intestine, and at 8, 20, and 36 h of colonic incubation (simulating the ascending, transverse, and descending colon). Samples for microbiological analysis were inoculated immediately after collection; the remaining aliquots were stored at −80 °C until further analysis. Digestive results are reported as concentrations within each bioreactor vessel, without correction for the progressive dilution of the initial sample (5.8-fold in the stomach, 8.3-fold in the small intestine, and 10.9-fold in the colon).

### 2.8. Analysis of the Digestion Extract

Following the simulated gastrointestinal digestion (Section 2.7), aliquots were periodically sampled from the bioreactors for subsequent comprehensive chemical and microbiological characterization. For all chemical analyses, samples were centrifuged at 10,000× *g* for 10 min at 4 °C to remove insoluble particulate matter and then filtered through 0.45 μm nylon membranes. Samples for microbiological analysis were processed without prior filtration or centrifugation.

Total phenolic compounds were quantified using the Folin–Ciocalteu colorimetric method, as detailed in Section 2.3.

Antioxidant capacity was determined by two spectrophotometric methods: the 2,2-diphenyl-1-picrylhydrazyl (DPPH) and the 2,2′-azinobis (3-ethylbenzothiazoline-6-sulfonic acid) (ABTS) radical scavenging assays, as described in Section 2.5.

Short-chain fatty acids (SCFAs) produced by the microorganisms after the different simulated colon stages and during growth on papaya extract were quantified by gas chromatography equipped with a split/splitless injector and a flame ionization detector (GC-FID; Shimadzu 2010 Plus, Kioto, Japón), following the procedure described by Rivas et al. [30]. Concentrations were determined from the ratio of the analyte peak area to that of the internal standard (2-ethylbutyric acid).

Quantification of viable microorganisms: 10 mL aliquots of each sample were aseptically diluted in 90 mL of 1% peptone water (Condalab, Madrid, Spain) and homogenized for 2 min in a stomacher (Lab-Blender 400 Seward Lab., London, UK). Serial 10-fold dilutions were prepared, and 0.1 mL aliquots were plated onto specific agar media. Total viable bacteria were enumerated on plate count agar (PCA; Condalab) after 48 h at 30 °C. Lactic acid bacteria (LAB) were counted on MRS agar (Condalab), acidified to pH 5.6 with 10% acetic acid, and incubated for 48 h at 30 °C. Enterococci were enumerated on Slanetz and Bartley agar (SB; Condalab) at 37 °C for 48 h. Enterobacteria were counted on violet red bile glucose agar (VRBG; Condalab) after 24 h at 30 °C. Microbial counts were expressed as log CFU/mL.

### 2.9. Statistical Analysis

Statistical analysis of the data was performed using SPSS for Windows, version 21.0 (IBM Corp., Armonk, NY, USA). Descriptive statistics were determined. For parameters related to general bioactivity and chemical composition, differences within and between groups were assessed using one, two, and three-way analysis of variance (ANOVA), and averages were separated using Tukey’s Honestly Significant Difference test (*p* ≤ 0.05). For the simulated digestion experiment, differences between extract and control samples across digestion stages were assessed using repeated-measures ANOVA, and Bonferroni-corrected post hoc tests were applied, with significance set at *p* ≤ 0.05. Additionally, Principal Component Analysis (PCA) was conducted on the correlation matrix of the variables.

## 3. Results and Discussion

### 3.1. Phenolic Compounds of Extracts Obtained from the Peels and Seeds of Different Papaya Varieties

Table 1 presents the total phenolic compounds (mg GAE/100 g dry sample) extracted from different papaya varieties.

The mean total phenolic content (TPC) in the papaya peel ranged from 374.01 to 511.09 mg GAE/100 g dry weight. No significant differences (*p* ≥ 0.05) were found in TPC among the different papaya varieties for the peel. In contrast, the mean TPC of the seeds varied significantly among varieties, ranging from 229.61 to 1029.24 mg GAE/100 g dry weight (*p* ≤ 0.05). The seeds of the Maradol variety had a TPC of 1029.24 mg GAE/100 g dry weight, which was much higher than the TPC of any of the other varieties tested (Table 1).

Significant differences in total phenolic content (TPC) were observed between by-products within the same variety. In the Maradol variety, the TPC was significantly higher in the seeds (1029.24 mg GAE/100 g) than in the peel (467.85 mg GAE/100 g). Conversely, in the Tainung variety, the peel had a significantly higher TPC (511.09 mg GAE/100 g) than the seeds (229.61 mg GAE/100 g).

In similar studies, papaya by-products have been reported to contain high phenolic concentrations. Their phenolic content is sensitive to variety, ripening stage, and extraction method; for peels, values typically fall around ~200–900 mg GAE/100 g dry weight under conventional ethanolic extraction and can exceed 1500 mg/100 g when phenolic-enriching or fractionation protocols are used [12]. For seeds, the literature values under conventional or optimized ethanolic extractions commonly range from ~250 to ~600 mg GAE/100 g dry weight [13], and increases with ripening have been observed [31]. In line with these trends, process-optimization studies demonstrate that extraction parameters markedly modulate seed TPC, underscoring the joint influence of methodological factors and developmental stage.

The results of the identification and relative abundance of bioactive compounds in the papaya extracts, derived from HPLC-ESI-QTOF analysis, are presented in Table 2 and Table 3. Table 2 systematically categorizes the identified constituents. Included for each entry are their respective retention times (Rt) and the characteristic fragments acquired during MSn mass spectrometry experiments.

Identification (Table 2) and relative abundance (Table 3) of bioactive compounds by HPLC-ESI-QTOF in papaya by-products (peel vs. seed, and between varieties) reveal several interesting patterns. First, phenolic compounds (peaks 1–7) show a notably higher relative abundance in seed extracts compared to peel extracts for many peaks, whereas acids (peaks 8, 9, 10) and flavonoids (peaks 11, 12) show mixed results. While the by-product factor (seed vs. peel) was significant for most analytes (*Pb* < 0.001), the variety factor was not (*Pe* > 0.05). This indicates conserved individual-peak profiles across varieties despite differences in total phenolics.

This compositional data supports the TPC results, where seeds of certain varieties (e.g., Maradol) had much higher total phenolic content than peels, consistent with seeds being a richer source of individual phenolics and flavonoids. Several literature reports align with these findings. A recent work by Alonso et al. [43] evaluated the phenolic profile of papaya by-products (including seeds and peel) using HPLC-MS/MS and reported that seeds generally contained higher levels of phenolic compounds compared to the peel extracts. Similarly, Robles-Apodaca et al. [13] optimized extraction from papaya seeds and showed that seed extracts under optimized conditions are enriched in phenolics and flavonoids, contributing to high antioxidant activity.

These observations (Table 2 and Table 3) suggest that the seed represents a major reservoir of phenolic compounds in papaya by-products, particularly in certain varieties. In addition, the identification of compounds such as caffeoylquinic acid derivatives (peak 7), isorhamnetin dihexoside (peak 11), and quercetin-3-O-rutinoside (peak 12) provides a more detailed understanding of the chemical composition of papaya by-products, allowing discrimination not only at the level of total phenolics but also among specific phenolic and flavonoid subclasses. Detection of caffeoylquinic acid derivatives and quercetin glycosides (including quercetin-3-O-rutinoside) aligns with prior papaya by-product profiles [5,44]; isorhamnetin glycosides have been reported in papaya tissues (leaves), supporting their occurrence within the species, although tissue-specific confirmation in peel and seed warrants verification [45].

### 3.2. Antioxidant Activity of Extracts Obtained from the Peels and Seeds of Different Papaya Varieties

The results of antioxidant activity, quantified via the 2,2-diphenyl-1-picrylhydrazyl (DPPH) radical scavenging assay and the 2,2′-azinobis-(3-ethylbenzothiazoline)-6-sulfonic acid (ABTS) radical scavenging assay, are presented in mg of Trolox per 100 g of extract in Table 4.

The mean antioxidant activity, as determined by the DPPH method, in papaya shells ranged from 189.08 to 287.91 mg Trolox/100 g dry weight. For the seeds, the mean values varied between 208.19 and 276.33 mg Trolox/100 g dry weight. No significant differences (*p* ≥ 0.05) in DPPH antioxidant activity were observed among the different papaya varieties, nor were there significant differences between the two by-products (shell and seeds) within the same variety.

Analysis of the ABTS method results revealed distinct patterns in antioxidant activity across papaya by-products. The mean values for papaya shell ranged from 144.49 to 209.73 mg Trolox/100 g dry weight, with significant differences (*p* ≤ 0.05) observed among the different varieties. For the seeds, mean ABTS values varied between 168.10 and 289.96 mg Trolox/100 g dry weight, and significant differences (*p* ≤ 0.05) were indeed found among varieties. Notably, the Maradol variety’s seeds (280.96 mg Trolox/100 g) exhibited significantly higher antioxidant activity compared to both the Solo (168.10 mg Trolox/100 g) and Tainung (191.74 mg Trolox/100 g) varieties. The Hawaiian variety (227.02 mg Trolox/100 g) did not show significant differences when compared to other varieties. Furthermore, significant differences in ABTS activity were also observed between the shell and seeds, specifically within the Maradol variety (Table 4).

These results are consistent with assay chemistry. ABTS detects both hydrophilic and lipophilic antioxidants, while DPPH performs best in non-polar media; as a result, DPPH often under-resolves differences that ABTS can detect in mixed plant matrices like papaya by-products. This helps explain why DPPH showed no significant variety or shell–seed effects, but ABTS did reveal variety effects in seeds [46,47].

Consistent with our results, in papaya, Jeon et al. [31] showed that seed extracts consistently exhibited higher DPPH and ABTS radical scavenging capacities than the pulp/shell at comparable concentrations, and that total phenolics were strongly correlated with antioxidant activity measured by the ABTS method. Similarly, and highlighting cultivar-specific responses, Gaye et al. [48], using aqueous extracts of three Senegalese cultivars, reported that by DPPH, seeds showed antioxidant activity greater than or equal to that of shells in all cultivars. In contrast, and reinforcing the assay-dependent sensitivity mentioned above, ABTS results were cultivar-dependent: in the first cultivar, seeds outperformed shells; in the second cultivar, shells outperformed seeds; and in the third cultivar, seeds again outperformed peels, underscoring genotype-tissue-assay interactions.

### 3.3. Antimicrobial Activity of Extracts Obtained from the Shells and Seeds of Different Papaya Varieties

For antimicrobial activity, analyses were performed against bacteria (*Bacillus cereus*, *Staphylococcus aureus*, *Salmonella choleraesuis*, *Listeria innocua*, and *Listeria monocytogenes*). Table 5 presents the antibacterial activity results (expressed as % inhibition) of the various papaya extracts (Solo, Hawaiian, Maradol and Tainung) at concentrations of 1000, 700, and 350 mg/L.

The variety of papaya significantly influenced the antimicrobial capacity of the extracts. The Hawaiian variety demonstrated superior inhibitory effects against four of the five bacterial strains tested: *Listeria innocua*, *Listeria monocytogenes*, *Bacillus cereus*, *Staphylococcus aureus*, and *Salmonella choleraesuis*. Overall, inhibition percentages ranged from 40.60% to 84.63% against *L. innocua*, 61.47% to 89.21% against *L. monocytogenes*, 52.92% to 86.17% against *B. cereus*, 52.90% to 92.64% against *S. aureus*, and 66.40% to 94.92% against *S. choleraesuis*.

Regarding the plant part, seed extracts consistently exhibited higher inhibition than shell across all five bacteria, and the by-product main effect was significant (*Pb* < 0.001). Although inhibition against *S. choleraesuis* was often high, the most responsive species depended on the variety (e.g., Maradol peaked for *L. innocua*; Tainung for *L. monocytogenes*). Overall, *S. aureus* and *L. monocytogenes* tended to show high inhibition, whereas *B. cereus* and *L. innocua* were more variable, with *L. innocua* lower in several varieties.

The antimicrobial activity was clearly influenced by extract concentration in both shell and seed extracts. At 1000 mg/L extract, inhibition reached ~91% for *L. monocytogenes* and remained in the high 70–80 % for the other species, decreasing stepwise at 700 and 350 mg/L (e.g., down to ~48% for *L. innocua* at 350 mg/L).

Significant interaction effects were detected for variety × by-product (Pe*b) across all five bacteria (*p* < 0.001). Variety × concentration (Pe*c) was also significant for all five. By-product × concentration (Pb*c) was non-significant only for *L. innocua* (*p* = 0.18) but significant for *L. monocytogenes*, *B. cereus*, and *S. aureus* (*p* < 0.001) and for *S. choleraesuis* (*p* = 0.05).

It is likely that the antimicrobial activity of the papaya extracts reflects a multi-compound mode of action, in which phenolic acids (e.g., caffeoylquinic) and flavonol glycosides (quercetin-based) contribute via membrane disruption, enzyme inhibition, and anti-biofilm/quorum-sensing effects [49,50,51]. In parallel, benzyl isothiocyanate (BITC)—formed from benzyl glucosinolate detected in our profile (Table 3)—shows potent antibacterial activity, notably against *L. monocytogenes* and *S. aureus*, as evidenced by recent studies in food-relevant systems and mechanistic assays [52,53]. Overall, this evidence supports interpreting the observed inhibition as the integrated outcome of multiple bioactives rather than a single dominant constituent.

These results are consistent with reports indicating that papaya by-products possess broad in vitro antibacterial activity, modulated by cultivar, plant part, and extraction conditions. As comparable examples, first, methanolic extracts of cv. Sekaki/Hong Kong seeds strongly suppressed *B. cereus* and *Salmonella enterica*, illustrating both the higher mean seed activity and solvent dependence [54]; second, an ethanolic extract of papaya shell reached a minimum inhibitory concentration of ≈1.56 mg/mL against *L. monocytogenes*, in the same order of magnitude as the highest dose used here (1000 mg/L) and consistent with ≈90% inhibition at that level [55]. Broader evaluations of tropical fruit by-product extracts corroborate dose-dependent suppression of *Listeria*, *Salmonella*, *Bacillus*, and *Staphylococcus*, reinforcing the main effect of concentration and the significant variety x concentration interaction observed in our study [56]. Taken together, these findings support and justify the results obtained in the present study, highlighting that both the concentration effect and varietal differences observed here are consistent with previous literature.

### 3.4. Characterisation of Digestion Extracts

Table 6 shows the evolution of total phenolic compounds (TPCs) and antioxidant capacity (determined by ABTS and DPPH methods) of papaya extracts during in vitro simulated gastrointestinal digestion. The digestion phases analyzed included the initial sample, stomach, small intestine, and colon simulation. For the colon phase, results are presented as a combined value, as no significant effect on phenolic content or antioxidant activity was observed across its distinct stages.

The TPC content of the papaya extract was significantly higher (*p* ≤ 0.05) than the control in the initial sample (207.00 mg GAE/L vs. 27.75 mg GAE/L), and this significant difference was maintained through the stomach and small intestine phases (Table 6). This sustained higher TPC content suggests a notable bioaccessibility and stability of papaya phenolics against gastric acidity and small intestinal enzymatic degradation, indicating their potential availability for absorption in the upper gastrointestinal tract [57,58]. In the colon simulation, no significant difference in TPC was observed (81.74 vs. 64.89 mg GAE/L), which may reflect a combination of upper-tract absorption, microbial transformation into non-Folin-reactive products, and the assay’s limited specificity for structurally modified phenolics [59,60].

Regarding antioxidant capacity, papaya extracts showed higher ABTS than the control in the initial and stomach phases, and a significant difference also appeared in the small intestine, although the control showed the higher absolute value in that phase (0.92 vs. 0.81 mg Trolox/L). In the colon simulation, ABTS remained elevated in papaya relative to the control. Conversely, DPPH showed a significant effect for papaya only during the stomach and small intestine (Table 6), becoming negligible in the colon, consistent with its lower discriminative power in aqueous/mixed media. Furthermore, during simulated digestion, TPC correlated with DPPH antioxidant activity across different digestion phases, but not with that of ABTS. This assay-dependent difference is consistent with previous reports showing method-specific behavior in foods and during in vitro digestion [61,62].

Papaya by-products contain numerous phenolic compounds that trigger a potent radical scavenging response [5,12]. Our results from the digestion assay are consistent with digestion studies on tropical fruits. In an in vitro model, antioxidant activity, measured using two complementary assays, increased in some fruits, whereas it remained virtually stable in papaya after digestion, supporting the persistence of activity observed in this study [63]. Additionally, similar behavior has been reported elsewhere: in passion fruit peel, antioxidant activity remained relatively strong after simulated digestion [64]. In papaya systems, a study on fermented papaya purée showed increased post-digestion bioaccessibility of phenolics with sustained antioxidant readouts compared with the undigested control [65].

Table 7 displays the production of short-chain fatty acids (SCFAs) exclusively during the simulated colon phase, which is consistent with their established microbial origin.

The papaya extracts significantly enhanced acetic acid content across all three colon stages. Notably, in the proximal colon, acetic acid in the papaya extract sample (1.43 g/L) was approximately 1.59 times higher than in the control (0.90 g/L). Similarly, in the transverse colon, it was 1.53 g/L, approximately 1.37 times higher than the control (1.11 g/L). Propionic acid also followed this stimulatory trend, being 1.46 times higher (1.85 g/L) in the proximal colon and 1.14 times higher (1.89 g/L) in the transverse colon in the papaya extract digester compared to the control (1.46 g/L and 1.65 g/L, respectively). These increases underscore the potential of papaya by-product extract as a fermentable substrate for the gut microbiota, promoting the production of these key SCFAs beneficial for gut health. In contrast, butyric acid content did not show significant differences between the papaya extract and control at any stage (Table 7), suggesting the extract’s components preferentially stimulate acetate and propionate-producing bacteria or do not favor butyrate synthesis under these conditions. Other SCFAs (valeric, isovaleric, caproic, isobutyric, and isocaproic acids) were not detected above the 0.01 g/L quantification limit.

Compared to the control group, the papaya phenolic extract favoured acetate and propionate over butyrate under these conditions. This pattern aligns with polyphenol-mediated modulation of microbial fermentation, which often shifts metabolism toward acetate and propionate. By contrast, butyrate responses are context dependent and vary with substrate availability, inoculum composition, pH, and transit time [66,67]. Notably, the cited red wine and cocoa trials reported microbiota changes without direct SCFA quantification [68,69]; thus, our SCFA pattern aligns with biological plausibility rather than direct external confirmation. Complementary in vitro studies indicate that polyphenols can influence fermentation, promoting the production of acetate and propionate [70].

The dynamics of key microbial populations are compiled in Table 8, with initial detectable counts in the small intestine and highest counts consistently in the proximal colon, reflecting the microbial load from the inoculum and subsequent resource utilization. The papaya extract influenced microbial populations variably. Total viable aerobic mesophilic bacteria showed significantly higher counts in the small intestine for the papaya extract group (8.52 log CFU/mL vs. control: 7.48 log CFU/mL), and significantly in the transverse colon (8.64 log CFU/mL vs. control: 8.37 log CFU/mL). Enterobacteriaceae counts also increased significantly in the proximal colon with papaya; as a facultative, heterogeneous group, this rise likely indicates greater resource use rather than a uniformly beneficial shift.

Conversely, lactic acid bacteria (LAB) were detected only in the colon and exhibited significantly higher counts in the control sample across all colon stages (Table 8). Similarly, enterococci counts were significantly higher in the control only in the distal colon (4.98 log CFU/mL vs. papaya: 4.97 log CFU/mL). The concurrent rise in total aerobes (small intestine, transverse colon) and Enterobacteriaceae in the proximal colon, together with the significant declines in LAB and enterococci, is consistent with selective inhibition by papaya bioactives. Dietary polyphenols often exert stronger antimicrobial activity against Gram-positive taxa—compatible with the observed LAB/enterococci decreases—while some Gram-negative facultatives tolerate and/or metabolize polyphenol-derived substrates [71,72]. In addition, papaya seeds provide benzyl isothiocyanate (BITC), a broad-spectrum antimicrobial with documented potency against Gram-positive bacteria that could reinforce this pattern [73,74].

The stage-resolved increases in total viable aerobes (small intestine, transverse colon) and the higher proximal-colon counts align with how phenolic-rich substrates shape gut ecology: phenolics from papaya by-products supply bioactive molecules that are transformed by microbes, supporting growth of facultative and saccharolytic groups in early/upper compartments and sustaining high loads proximally as fermentable inputs are most available there [75]. Notably, the effect appears to be source-specific and may depend on the polyphenol class, the co-presence of fermentable dietary fibers, and the colon region, with certain polyphenols preferentially favoring acetate production while others more strongly influence butyrate or propionate [76]. This interpretation is consistent with evidence that tropical fruit-derived phenolics (including papaya peels) are abundant and bioactive, and that polyphenol-rich matrices can remodel communities and fermentation readouts in vitro [5,70].

### 3.5. Multivariate Analysis

Figure 1 displays the Principal Component Analysis (PCA), which projects the samples and variables onto the plane defined by Principal Component 1 (PC1) and Principal Component 2 (PC2). These two components account for a combined 74.98% of the total variance (PC1 at 48.79% and PC2 at 26.19%). PC1 is primarily associated with antimicrobial activity, and it is located on the right side of the plot. This is where varieties like Hawaiian are positioned and seeded, suggesting a higher antimicrobial capacity in their extracts. Moreover, the position of phenolic compounds such as Hidroxybenzoic acid isomer (peak 2), Quercetin-3-Orhamnosyl rutinoside (peak 3), and Zeatin-9-glucoside (peak 6) suggests their potential contribution to the antimicrobial activity observed in these extracts.

On the other hand, PC2 differentiates the samples based on the antioxidant activity, with the variety Maradol showing a positive correlation with this activity. The variety Tainung exhibits a higher concentration of compounds such as Protocatechuicacid-O-hexoside (peak 10), which may be involved in the inhibition of bacterial growth.

The PCA clearly visualized data dispersion, effectively separating the extracts based on two orthogonal bioactivity axes: antimicrobial capacity (PC1) and antioxidant potential (PC2). This statistical separation supports the specialized functional profiles of the papaya by-products. The positive correlation along PC1 between antimicrobial assays and Hawaiian seed extracts provides a strong statistical basis for their superior inhibitory performance, chemically linked to hydroxybenzoic acid isomer and Quercetin-3-O-rhamnosyl rutinoside. Conversely, the differentiation along PC2 highlights the compositional uniqueness of the Maradol variety, which is aligned with high antioxidant activity, confirming its superior antioxidant profile. These results offer a rationale for a nuanced valorization strategy: Hawaiian extracts are the most promising candidates for strong antimicrobial applications, while Maradol extracts are better suited for use as natural antioxidants, a targeted approach central to optimizing the use of these by-products in the food industry.

## 4. Conclusions

In conclusion, Maradol seeds displayed the highest total phenolic content and antioxidant values, evidencing superior compositional and functional profiles. Seed extracts surpassed shell extracts in antibacterial efficacy across the tested panel, with Hawaiian seed extracts exhibiting the greatest overall inhibition. Under simulated gastrointestinal conditions, combined shell and seed extracts preserved phenolics across the gastric and intestinal phases. During colonic fermentation, they enhanced acetate and propionate production. These findings support the functional valorization of papaya by-products and indicate that selecting extracts by bioactive profile can optimize applications in the food industry. While the dynamic in vitro digestion model used is robust and considered state-of-the-art, it still has inherent limitations and may not fully recapitulate in vivo physiology. Future studies should validate these findings in vivo to more clearly elucidate microbiome-mediated mechanisms and dose–response relationships.

## Figures and Tables

**Figure 1 foods-14-03885-f001:**
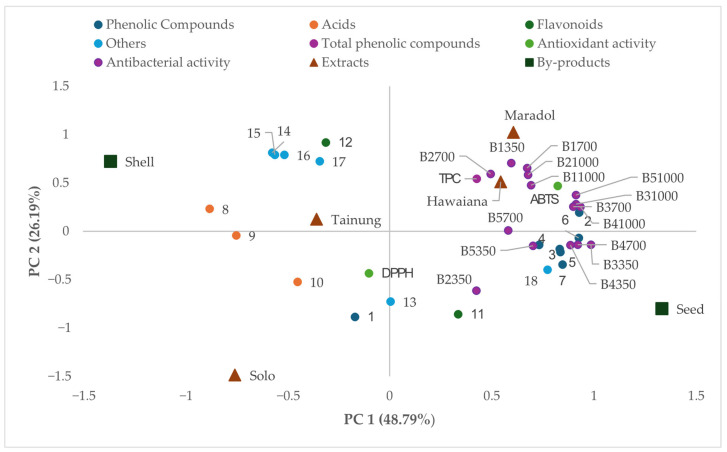
Plane defined by principal components 1 and 2 (PC1 and PC2) showing the parameters and compounds analysed (TPC: total phenolic compounds; antioxidant activity: DPPH and ABTS; antimicrobial activity at concentration different (1000, 700, 350 ppm) against *L. innocua* (B1), *L. monocytogenes* (B2), *B. cereus* (B3), *S. aureus* (B4) and *S. choleraesuis* (B5). The numerical code corresponds to the compounds listed in Table 2; Phenolic Compounds (1–7); Acids (8–10); Flavonoids (11–12); Others (13–18). The proximity between variables (activity vectors and compounds) and samples (extracts) indicates a positive correlation or similarity in profile.

**Table 1 foods-14-03885-t001:** Total phenolic compounds (mg GAE/100 g dry sample) of the extracts obtained from different papaya varieties.

Extract	Shell			Seeds			
	Mean		SD *	Mean		SD	*p* Values
Solo	374.01	±	83.89	371.23	±	17.41 ^b^	0.950
Hawaiana	505.82	±	22.07 _1_	355.76	±	75.90 ^b^_2_	0.030
Maradol	467.85	±	46.02 _2_	1029.24	±	37.13 ^a^_1_	<0.001
Tainung	511.09	±	85.14 _1_	229.61	±	30.74 ^b^_2_	0.001
*p values*	0.104			<0.001			

* SD: standard deviation. Superscripts (a,b): among varieties within the same by-product (*p* ≤ 0.05). Subscripts (1,2): between by-products (peel vs. seed) within the same variety (*p* ≤ 0.05).

**Table 2 foods-14-03885-t002:** Identification of the main compounds from papaya by-products extracts analyzed by HPLC-ESI-QTOF.

Peak	Rt (min)	[M-H]^−^	MS/MS (*m*/*z)*	Compounds Identified
** *Phenolic Compounds* **				
1 *	4.231	341	179; 135; 101	Caffeic acid O-glucoside ^1,2^
2 **	5.705	167	124	Vanillic acid ^3^
3 **	11.11	137		Hidroxybenzoic acid isomer 1 ^4^
4 **	11.86	137		Hidroxybenzoic acid isomer 2 ^4^
5 **	12.69	137		Hidroxybenzoic acid isomer 3 ^4^
6 **	13.6	137		Hidroxybenzoic acid isomer 4 ^4^
7 ****	15.6	677.5	250; 132	Caffeoylquinic acid derivative ^5^
** *Acids* **				
8 **	4.611	175	115	Ascorbic acid ^3,6^
9 *	5.539	191	173; 127	Citric acid ^1,7^
10 **	13.83	315	108; 152	Protocatechuic acid-O-hexoside ^7,8^
** *Flavonoids* **				
11 ****	13.5	639	167; 153	Isorhamnetin dihexoxide derivative ^9^
12 *	14.74	755	300; 581	Quercetin-3-Orhamnosyl rutinoside ^1,3,7^
** *Others* **				
13	3.735	129	103	Unknown
14 ***	3.964	131	113; 114	L-asparagine ^1,10,11,12^
15 ***	13.75	380	101	Zeatin-9-glucoside ^13^
16	14	254	128	Unknown
17 *	14.35	408	259; 166	Benzyl glucosinolate ^1,3^
18	15.36	236	120; 192	Unknown ^4^

^1^ MassBank, [25,26]; ^2^ Rivera et al. [32]; ^3^ Kafuko, [33]; ^4^ Panusa et al. [10]; ^5^ Kramberger et al. [34]; ^6^ Gayosso-García et al., [35]; ^7^ Spínola et al. [36], ^8^ Quifer-Rada et al. [37]; ^9^ Lin et al. [38]; ^10^ Liu et al. [39]; ^11^ Gogna et al. [40]; ^12^ Kahsay et al. [41]; ^13^ Mihai et al. [42]. * Reported spectrum in papaya in bibliography and corroborated in massbank. ** Reported spectrum in bibliography. *** Reported compound in bibliography and corroborated in massbank. **** Reported spectrum in other foods in bibliography.

**Table 3 foods-14-03885-t003:** Relative abundances in arbitrary area units of the main compounds from papaya by-products extracts analyzed by HPLC-ESI-QTOF.

Factors	Extracts (E)				By-Products (B)		*p*-Values	
Levels	Hawaiian	Tainung	Maradol	Solo	Peel	Seed	*Pe*	*Pb*
** *Phenolic Compounds* **								
1	7.38	7.56	14.45	6.77	8.06	10.02	0.732	0.710
2	10.61	9.87	2.02	9.10	2.41	13.39	0.748	<0.001
3	51.00	152.45	72.45	60.32	0.65	167.45	0.858	<0.001
4	74.80	456.29	198.40	120.25	0.91	423.96	0.739	<0.001
5	251.79	709.97	361.20	290.55	<0.1 *	806.75	0.875	<0.001
6	30.91	58.82	22.30	18.69	0.43	64.93	0.854	<0.001
7	66.49	75.20	69.57	68.35	58.51	81.30	0.993	0.220
** *Acids* **								
8	13.51	19.65	22.98	15.01	30.01	5.56	0.952	<0.001
9	25.33	35.71	39.40	29.40	39.24	25.68	0.627	<0.001
10	26.76	43.04	45.55	65.02	45.66	44.52	0.778	0.963
** *Flavonoids* **								
11	19.77	9.87	56.99	48.02	<0.1 *	67.33	0.803	<0.001
12	117.66	106.74	0.24	72.23	147.04	1.39	0.772	<0.001
** *Others* **								
13	149.10	222.95	268.79	210.97	186.54	239.36	0.371	0.274
14	228.28	237.26	149.18	242.25	311.81	116.68	0.911	<0.001
15	15.24	13.40	4.79	17.17	23.15	2.15	0.877	<0.001
16	20.20	40.50	4.98	45.98	54.50	1.33	0.784	<0.001
17	57.82	190.74	13.01	231.19	201.97	44.40	0.490	0.144
18	19.11	20.80	18.72	28.29	5.12	38.33	0.972	<0.001

The numerical code corresponds to the compounds listed in Table 2. * The limit of detection was 0.1 mg/L.

**Table 4 foods-14-03885-t004:** Antioxidant activity determined by DPPH and ABTS methods (mg Trolox/100 g) of extracts obtained from by-products of different papaya varieties.

DPPH								ABTS						
Extracts	Shell			Seeds				Shell			Seeds			
	Mean		SD *	Mean		SD	*p* Values	Mean		SD	Mean		SD	*p* Values
Solo	218.79	±	11.38	271.40	±	46.92	0.132	144.49	±	4.75 ^c^	168.10	±	3.68 ^b^	0.317
Hawaiana	189.08	±	82.40	255.87	±	82.38	0.377	172.19	±	7.27 ^b^	227.02	±	93.61 ^a,b^	0.366
Maradol	226.95	±	48.54	208.19	±	117.20	0.387	166.42	±	6.61 ^b^_2_	280.96	±	15.04 ^a^_1_	0.001
Tainung	287.91	±	33.25	276.33	±	30.96	0.603	209.73	±	4.52 ^a^_1_	191.74 ^b^	±	7.86 ^b^_2_	0.002
*p values*	0.194			0.499				<0.001			0.033			

* SD: standard deviation. Superscripts (a,b,c): among varieties within the same by-product (*p* ≤ 0.05). Subscripts (1,2): between by-products within the same variety (*p* ≤ 0.05).

**Table 5 foods-14-03885-t005:** Antibacterial activity against different bacteria (% inhibition) of the different extracts obtained from the by-products of different papaya varieties.

	*L. innocua*			*L. monocytogenes*			*B. cereus*			*S. aureus*			*S. choleraesuis*		
	Mean		SD *	Mean		SD	Mean		SD	Mean		SD	Mean		SD
*Extract (E)*															
Solo	40.60 ^c^	±	33.15	61.47 ^c^	±	37.08	52.92 ^b^	±	46.52	52.90 ^c^	±	47.09	69.03 ^c^	±	31.98
Hawaiian	62.26 ^b^	±	39.83	89.21 ^a^	±	25.19	86.17 ^a^	±	32.26	92.64 ^a^	±	13.02	94.92 ^a^	±	7.76
Maradol	84.63 ^a^	±	17.02	74.54 ^b^	±	28.17	82.69 ^a^	±	21.96	69.47 ^b^	±	37.84	77.79 ^b^	±	32.76
Tainung	59.74 ^b^	±	43.47	88.45 ^a^	±	17.47	53.72 ^b^	±	47.13	56.20 ^c^	±	42.53	66.40 ^c^	±	36.38
*By-products (B)*															
Shell	54.73	±	40.78	68.47	±	34.46	50.82	±	43.96	41.28	±	40.37	67.09	±	33.07
Seed	68.88	±	32.68	88.37	±	19.05	86.93	±	26.86	94.32	±	10.18	86.98	±	24.97
*Concentration (C)* (mg/L)															
1000	76.78 ^a^	±	31.12	90.90 ^a^	±	23.97	81.68 ^a^	±	32.40	79.68 ^a^	±	29.96	87.65 ^a^	±	22.12
700	60.42 ^b^	±	37.42	85.02 ^b^	±	25.61	71.29 ^b^	±	42.19	64.67 ^b^	±	42.80	73.79 ^b^	±	34.20
350	48.22 ^c^	±	39.25	59.33 ^c^	±	29.37	53.66 ^c^	±	43.06	59.06 ^b^	±	44.00	69.65 ^c^	±	33.19
*p* Values															
Pe	<0.001			<0.001			<0.001			<0.001			<0.001		
Pb	<0.001			<0.001			<0.001			<0.001			<0.001		
Pc	<0.001			<0.001			<0.001			<0.001			<0.001		
Pe*b	<0.001			<0.001			<0.001			<0.001			<0.001		
Pe*c	<0.001			<0.001			<0.001			0.03			<0.001		
Pb*c	0.18			<0.001			<0.001			<0.001			0.05		

* SD: standard deviation. Superscripts (a,b,c): indicate statistical differences (*p* ≤ 0.05) between factors.

**Table 6 foods-14-03885-t006:** Total phenolic compounds (mg GAE/L) and antioxidant capacity (ABTS and DPPH methods) (mg Trolox/L) of papaya extracts during simulated in vitro gastrointestinal digestion.

		Initial Sample			Stomach Simulation			Small Intestine Simulation			Colon Simulation		
Parameter	Sample	Mean		SD	Mean		SD	Mean		SD	Mean		SD
TPC	Control	27.75	±	3.19	21.15	±	2.12	18.89	±	1.61	64.89	±	14.35
	Papaya	207.00	±	12.65 *	318.23	±	8.79 *	149.28	±	8.76 *	81.74	±	13.47
	*p values*	<0.001			<0.001			<0.001			0.150		
ABTS	Control	0.43	±	0.01	0.48	±	0.02	0.92	±	0.02	0.62	±	0.10
	Papaya	0.76	±	0.01 *	0.73	±	0.02 *	0.81	±	0.01 *	0.85	±	0.12 *
	*p values*	0.001			0.006			0.017			0.004		
DPPH	Control	0.21	±	0.01	0.21	±	0.01	0.00	±	0.00	0.00	±	0.00
	Papaya	0.19	±	0.03	0.45	±	0.02 *	0.13	±	0.04 *	0.00	±	0.00
	*p values*	0.472			0.004			0.035			-		

TPC: total phenolic compounds; SD: standard deviation; Mean values with * are significantly different (*p* ≤ 0.05) compared to the control.

**Table 7 foods-14-03885-t007:** SCFA (g/L) production from papaya extracts during simulated in vitro gastrointestinal digestion.

		Colon Simulation (Stage)								
		Proximal			Transverse			Distal		
SCFAs	Sample	Mean		SD	Mean		SD	Mean		SD
Acetic acid	Control	0.90	±	0.16	1.11	±	0.01	1.80	±	0.00
	Papaya	1.43	±	0.00 *	1.53	±	0.00 *	1.83	±	0.00 *
	*p values*	0.041			<0.001			0.01		
Propionic acid	Control	1.46	±	0.01	1.65	±	0.01	1.90	±	0.00
	Papaya	1.85	±	0.01 *	1.89	±	0.01 *	1.96	±	0.00 *
	*p values*	<0.001			0.003			<0.001		
Butiric acid	Control	1.22	±	0.10	0.66	±	0.01	0.69	±	0.02
	Papaya	1.28	±	0.02	0.57	±	0.03	0.55	±	0.05
	*p values*	0.468			0.059			0.071		

SD: standard deviation. Mean values with * are significantly different (*p* ≤ 0.05) compared to the control.

**Table 8 foods-14-03885-t008:** Main microbial population counts (CFU/mL) from papaya extracts during simulated in vitro gastrointestinal digestion.

		Small Intestine Simulation			Colon Simulation (Stage)								
Microbial Population					Proximal			Transverse			Distal		
	Sample	Mean		SD	Mean		SD	Mean		SD	Mean		SD
Total viable bacteria	Control	7.48	±	0.02	8.79	±	0.08	8.37	±	0.04	8.31	±	0.06
	Papaya	8.52	±	0.03 *	9.04	±	0.16	8.64	±	0.13 *	8.34	±	0.05
	*p values*	<0.001			0.080			0.025			0.586		
Enterobacteria	Control	7.49	±	0.12	8.74	±	0.11	8.40	±	0.13	8.43	±	0.09
	Papaya	7.51	±	0.07	9.06	±	0.05 *	8.62	±	0.08	8.34	±	0.08
	*p values*	0.875			0.009			0.058			0.272		
Lactic acid bacteria	Control	0.00	±	0.00	6.35	±	0.06	6.33	±	0.05	6.37	±	0.01
	Papaya	0.00	±	0.00	6.27	±	0.19	5.63	±	0.01 *	4.96	±	0.06 *
	*p values*	-			0.486			<0.001			<0.001		
*Enterococci*	Control	0.00	±	0.00	6.52	±	0.04	6.50	±	0.18	6.36	±	0.16
	Papaya	0.00	±	0.00	6.39	±	0.13	5.97	±	0.34	4.98	±	0.07 *
	*p values*	-			0.173			0.079			<0.001		

SD: standard deviation. Mean values with * are significantly different (*p* ≤ 0.05) compared to the control.

## Data Availability

The original contributions presented in the study are included in the article, further inquiries can be directed to the corresponding author.

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
