# Peer review of "Valorization of Papaya By-Products: Bioactive Potential of Peel and Seeds and Their In Vitro Bioavailability"

_foods, 2025, doi:10.3390/foods14223885_

Round 1
Reviewer 1 Report
Comments and Suggestions for Authors
The manuscript under review represents an interesting study about the valorization of papaya fruit and its byproducts, as well as their potential biological effects.
Overall, the study is concordant to the journal's scope. However, it requires significant improvements. In general terms, the obtained results should be reviewed, the presentation of the results should be improved, and the discussions should be also improved providing concrete data for comparing the results. Critical input from the authors is also needed.
Some specific comments:
- The introduction lacks the references from the first general paragraphs (lines 37 and 41 of the first page).
- Furthermore, in the second paragraph of page 2, more information is needed regarding how papaya composition varies with the different factors indicated. Are these profiles of phenolic compounds? Their concentrations?
- In the methodology, in section 2.1, the sample collection points should be georeferenced. In addition, the quantities collected, as well as the transport and storage conditions, should be also included.
- On page 3 (lines 103 to 106), information regarding the fundaments of the methodology should be included in the discussion, not in the methodology. The method reference should also be included.
- In section 2.4 (line 120), C18, 18 should be subscripted.
- In section 3.1, GAE should be defined.
- In the results in Table 1, please review the SDs, as they are too high for this type of analysis. It is inconsistent that in some cases they reach almost 20%.
- The discussion in section 3.1 should be further elaborated. It is necessary to incorporate values reported for total phenol concentrations by other authors to understand their relationship with the values obtained.
- Table 2 indicates that an HPLC-UV-ESI-MS/MS analysis was used, but the methodology only mentions the Q-TOF detector. Please provide information about the UV detector. Was it UV or DAD? If DAD, also include the absorption peak wavelengths for each compound in the table.
- In Table 2, the identification of pick 7 as a derivative of caffeoylquinic acid is strange. This compound generally exhibits product ions of m/z 191, 179, and 135. Please attach MS and MS/MS spectra and a UV spectra to confirm its identity. Likewise, in pick 11, the non-detection of the base pick of isorhamnetin at m/z 314 is strange. Please also include the spectra to confirm this.
- I do not understand what is being reported in Table 3. Please replace this table with one that reports the concentrations with the error for each of the detected compounds. Also include information on the standards used for each quantification, the wavelength at which the quantification was performed, and the corresponding analytical parameters (equation of the line, R2, detection and quantification limits, linear range, etc.). Likewise, the results section of this table should be completely rewritten considering this information, as well as its discussion.
- In Table 4, the errors obtained should be reviewed; it is inconsistent to report errors of more than 50%. The statistical information should also be included. The description of the results should also be reviewed in detail, as there are values that do not match those in the table. In addition, the discussion section should be improved by considering values reported in other studies.
- Pay attention to scientific names; these should be italicized (section 3.3).
- In Table 6, the corresponding control should be indicated, and statistics should also be included. Is there any correlation between TPC and antioxidant activity? Please, expand on this information in the discussion.
- In section 3.5, a comprehensive and critical discussion should be included to close the work and lead to conclusions.
- The format of references should be improved; scientific names should be italicized.
Author Response
The manuscript under review represents an interesting study about the valorization of papaya fruit and its byproducts, as well as their potential biological effects.
Overall, the study is concordant to the journal's scope. However, it requires significant improvements. In general terms, the obtained results should be reviewed, the presentation of the results should be improved, and the discussions should be also improved providing concrete data for comparing the results. Critical input from the authors is also needed.
Some specific comments:
Comments 1: The introduction lacks references from the first general paragraphs (lines 37 and 41 of the first page).
Response 1: Thank you for the comments. We have added appropriate references to the first general paragraphs (formerly lines 37 and 41) covering global production/leading producers and Ecuador’s regional distribution.
Comments 2: Furthermore, in the second paragraph of page 2, more information is needed regarding how papaya composition varies with the different factors indicated. Are these profiles of phenolic compounds? Their concentrations?
Response 2: Thank you for the observation. We have expanded the second paragraph to clarify that the variability involves both phenolic profiles and concentrations.
Comments 3: In the methodology, in section 2.1, the sample collection points should be georeferenced. In addition, the quantities collected, as well as the transport and storage conditions, should be also included.
Response 3: Thank you for the observation. The requested information has been provided in Section 2.1: georeferenced sampling points (GPS, WGS84), quantities collected per variety, and transport and storage conditions.
Comments 4: On page 3 (lines 103 to 106), information regarding the fundaments of the methodology should be included in the discussion, not in the methodology. The method reference should also be included.
Response 4: Thank you for your comments. The information about the method foundations has been removed from the material and methods section, and the reference has been included.
Comments 5: In section 2.4 (line 120), C18, 18 should be subscripted.
Response 5: The error has been corrected
Comments 6: In section 3.1, GAE should be defined.
Response 6: Thank you for observation. GAE is defined in Section 2.3 (Materials and Methods).
Comments 7: In the results in Table 1, please review the SDs, as they are too high for this type of analysis. It is inconsistent that in some cases they reach almost 20%.
Response 7: We appreciate the reviewer’s observation regarding the standard deviations (SDs) presented in Table 1. However, we respectfully disagree with the assertion that SDs approaching 20% are inherently inconsistent or inappropriate for this type of analysis. These coefficients of variation remain within acceptable limits for biological assays.
It is important to consider that the determination of phenolic compounds in papaya varieties involves biological samples, which are naturally heterogeneous due to genetic, physiological, and environmental variability. Even within the same variety, factors such as ripening stage, cultivation conditions, and post-harvest handling can significantly influence phenolic content. This intrinsic variability is well-documented in the literature and often results in relatively high dispersion metrics (Vasco et al. 2008), even when working with a sample size of n = 12. In such contexts, SDs in the range of 10–20% are not uncommon and should not be interpreted as methodological inconsistency unless accompanied by evidence of procedural error or lack of reproducibility.
Vasco, C., Ruales, J., & Kamal-Eldin, A. (2008). Total phenolic compounds and antioxidant capacities of major fruits from Ecuador. Food Chemistry, 111(4), 816-823.
Comments 8: The discussion in section 3.1 should be further elaborated. It is necessary to incorporate values reported for total phenol concentrations by other authors to understand their relationship with the values obtained.
Response 8: Thank you for the constructive comment. We have revised Section 3.1 to further elaborate on the discussion and to benchmark our results against values reported by other authors.
Comments 9: Table 2 indicates that an HPLC-UV-ESI-MS/MS analysis was used, but the methodology only mentions the Q-TOF detector. Please provide information about the UV detector. Was it UV or DAD? If DAD, also include the absorption peak wavelengths for each compound in the table.
Response 9: We appreciate the reviewer’s attention to the methodological details. However, we respectfully clarify that the mention of HPLC-UV-ESI-MS/MS in Table 2 and results and discussion section was an inadvertent error. As correctly described in Section 2.4 of the Materials and Methods, the identification of bioactive compounds was performed exclusively using an HPLC system coupled to a quadrupole time-of-flight (Q-TOF) mass spectrometer (Agilent Model G6530), operating with an electrospray ionization (ESI) source in negative mode.
No UV or DAD detector was employed in this analysis. Compound identification was based solely on accurate mass measurements and fragmentation patterns obtained via tandem mass spectrometry, with provisional identities assigned through comparison with the MassBank database. Therefore, absorption peak wavelengths are not applicable to the data presented in Table 2.
We have corrected the terminology in the table and the result and discussion section to reflect the actual instrumentation used (HPLC-ESI-QTOF) to ensure consistency and accuracy throughout the manuscript.
Comments 10: In Table 2, the identification of pick 7 as a derivative of caffeoylquinic acid is strange. This compound generally exhibits product ions of m/z 191, 179, and 135. Please attach MS and MS/MS spectra and a UV spectra to confirm its identity. Likewise, in pick 11, the non-detection of the base pick of isorhamnetin at m/z 314 is strange. Please also include the spectra to confirm this.
Response 10: We appreciate the reviewer’s insightful observations regarding the identification of peaks 7 and 11 in Table 2. Indeed, the typical fragmentation pattern of caffeoylquinic acid derivatives includes product ions at m/z 191, 179, and 135, and the base peak of isorhamnetin is generally observed at m/z 314. We acknowledge that these are well-established markers in MS/MS analysis.
However, as indicated in the table caption and methodology, the identifications presented are tentative assignments based on spectral similarity to compounds previously reported in the literature (Kramberger et al., 2020; Lind et al., 2011) and matched against entries in the MassBank database. In the absence of authentic standards, these identifications are provisional and rely on high-resolution mass data and fragmentation patterns consistent with published spectra.
Regarding the UV spectra, we reiterate that no UV or DAD detector was used in this study, as clarified in our response to Comment 9.
Comments 11: I do not understand what is being reported in Table 3. Please replace this table with one that reports the concentrations with the error for each of the detected compounds. Also include information on the standards used for each quantification, the wavelength at which the quantification was performed, and the corresponding analytical parameters (equation of the line, R2, detection and quantification limits, linear range, etc.). Likewise, the results section of this table should be completely rewritten considering this information, as well as its discussion.
Response 11: We appreciate the reviewer’s detailed suggestions regarding Table 3 and acknowledge the importance of clarity and methodological rigor in reporting quantitative data. However, we respectfully clarify that the values presented in Table 3 correspond to arbitrary units of chromatographic peak area, not absolute concentrations. As such, these data do not derive from external calibration curves or quantification using analytical standards.
Due to the structural diversity and complexity of phenolic compounds in papaya and its by-products, obtaining pure standards for each detected compound was not feasible. This is a common challenge in plant metabolomics, particularly when dealing with minor or less commercially available phenolics.
On the other hand, the primary aim of this analysis was comparative, focusing on the relative abundance of phenolic compounds across different extracts and sub-products. The use of peak area (arbitrary units) allows for consistent intra-study comparisons, even in the absence of absolute quantification.
We agree that absolute quantification would enhance the analytical depth of the study. However, given the exploratory nature of this work and the constraints mentioned above, we opted for a semi-quantitative approach, which is widely accepted in metabolomic profiling when standards are unavailable.
To address the reviewer’s concern, we will revise the table caption and the corresponding section in the Results and Discussion to clearly state that the data are expressed in arbitrary units of peak area, and that compound identification was tentative and based on spectral matching with literature and databases. We will also remove any implication of absolute quantification and ensure that the comparative purpose of the table is explicitly stated.
Comments 12: In Table 4, the errors obtained should be reviewed; it is inconsistent to report errors of more than 50%. The statistical information should also be included. The description of the results should also be reviewed in detail, as there are values that do not match those in the table. In addition, the discussion section should be improved by considering values reported in other studies.
Response 12: We thank the reviewer for their careful evaluation of Table 4 and the associated discussion. We acknowledge the concern regarding the high variability observed in some of the reported values. However, as previously noted, the dispersion in antioxidant activity measurements—such as DPPH and ABTS—is a well-documented phenomenon in biological systems, particularly when dealing with secondary metabolites like phenolic compounds. These compounds are highly sensitive to environmental, genetic, and physiological factors, which can lead to substantial variability even within the same sample type.
The reported errors reflect this inherent biological variability and are consistent with findings in similar studies involving plant extracts and by-products. Nevertheless, we agree that the statistical reporting can be improved for clarity and rigor.
To address the reviewer’s concerns, we will incorporate p-values and explicitly indicate the post-hoc tests employed in the statistical analysis within the table footnotes. We will also carefully review and rectify any inconsistencies between the data presented in the tables and the corresponding descriptions in the Results section. Furthermore, the Discussion section will be expanded to contextualize our findings in relation to values reported in previous studies, emphasizing both similarities and differences, and providing potential biological and methodological explanations for the observed outcomes.
These revisions will enhance the transparency and interpretability of the data, while maintaining the integrity of the comparative analysis central to the study.
Comments 13: Pay attention to scientific names; these should be italicized (section 3.3).
Response 13: The error has been corrected
Comments 14: In Table 6, the corresponding control should be indicated, and statistics should also be included. Is there any correlation between TPC and antioxidant activity? Please, expand on this information in the discussion.
Response 14: Thank you for your comment. The control group used is already specified in Table 6. The exact p-values for each comparison have been included in the tables 6-8 following your indication, complementing the notation for statistical significance (p ≤ 0.05) that was already marked with an asterisk.
Regarding the correlation between TPC and antioxidant activity, yes, indeed, in our data, TPC correlates with antioxidant activity measured by DPPH across the digestion phases but shows no consistent relationship with ABTS. We have expanded the Discussion to explain these assay-dependent differences.
Comments 15: In section 3.5, a comprehensive and critical discussion should be included to close the work and lead to conclusions.
Response 15: We appreciate the reviewer's feedback. We have revised the manuscript to incorporate a comprehensive discussion in a new Section 3.5, which now serves to effectively close the results and lead into the conclusions.
Comments 16: The format of references should be improved; scientific names should be italicized.
Response 16: Thank you for your feedback. The errors have been corrected.

Reviewer 2 Report
Comments and Suggestions for Authors
Dear Authors
This study aimed to extract and characterize the bioactive compounds from the peel and seeds of different papaya varieties, evaluate their antioxidant and antimicrobial potential, as well as their behavior under simulated digestion
This article is very interesting and it encompasses many areas of research., but the presented methodology and the results in the tables are unclear; details are provided below.
Here are some comments about the article:
L.85-89 The choice of material for the research raises concerns; the four varieties come from different cultivations, with different agricultural conditions.
L.96 .... extracted from the byproducts... What do the authors mean by bioproducts? Is there an extract from different varieties for seeds and peel, or a blend? It is not clear?
Table 2. Were the same compounds identified in the seed and peel extract?
Table 2. Not “Caffeoylquinicacid” it should be “Caffeoylquinic acid” the same situation for “Protocatechuicacid” it should be “Protocatechuic acid”. Protocatechuic acid-O-hexoside it is Phenolic Compounds. Flavonoids belong to the group of phenolic compounds.
Table 3. Do the results relate to seed or peel extracts? By-products (seeds and peel) — are these residues from extraction? The same situation applies to tables 5
The Maradol variety contains a high content of polyphenols in the seeds (Table 1)- Is it a matter of the variety or the cultivation conditions?
Table 3. Are these the values of peak areas? Please provide the content of individual compounds in the papaya by-products extracts in the appropriate units, as well as the total polyphenols.
Table 6, 7,8 Extracts from papaya seeds or peel, from which variety?
Kind regards
Author Response
This study aimed to extract and characterize the bioactive compounds from the peel and seeds of different papaya varieties, evaluate their antioxidant and antimicrobial potential, as well as their behavior under simulated digestion
This article is very interesting, and it encompasses many areas of research, but the presented methodology and the results in the tables are unclear; details are provided below.
Here are some comments about the article:
Comments 1: L.85-89 The choice of material for the research raises concerns; the four varieties come from different cultivations, with different agricultural conditions.
Response 1: Thank you for pointing this out. Details of sourcing and agricultural conditions have been clarified in Materials and Methods (Section 2.1), including georeferenced collection sites, maturity at harvest, quantities by variety, and standardized transportation and storage procedures.
Comments 2: L.96 .... extracted from the byproducts... What do the authors mean by bioproducts? Is there an extract from different varieties for seeds and peel, or a blend? Is it not clear?
Response 2: Thank you for the comments. The information has been clarified in the Material and Methods section.
Comments 3: Table 2. Were the same compounds identified in the seed and peel extract?
Response 3: Table 2 lists the compounds identified in both seed and peel extracts. Table 3 reports the corresponding concentrations for each compound by-product (peel vs. seed) and variety.
Comments 4: Table 2. Not “Caffeoylquinicacid” should be “Caffeoylquinic acid” the same situation as “Protocatechuicacid” it should be “Protocatechuic acid”. Protocatechuic acid-O-hexoside it is Phenolic Compounds. Flavonoids belong to the group of phenolic compounds.
Response 4: The error has been corrected
Comments 5: Table 3. Do the results relate to seed or peel extracts? By-products (seeds and peel) — are these residues from extraction? The same situation applies to tables 5
Response 5: We confirm that the data presented in these tables are derived from the extracts prepared from the papaya by-products (peel and seed), not from any post-extraction residues. The term 'By-products (B)' in these tables simply denotes the raw plant material (peel or seed) that was subject to the extraction process.
Comments 6: The Maradol variety contains a high content of polyphenols in the seeds (Table 1)- Is it a matter of the variety or the cultivation conditions?
Response 6: Since Maradol, Tainung, and Solo were sampled at the same site (Paján, Manabí; n=12 per variety) and handled under identical postharvest conditions, the higher polyphenol content in Maradol seeds is more likely due to a varietal (genetic) effect than to growing conditions. Site effects primarily confound comparisons with the Hawaiian variety (from Manta). While small residual influences (inter-tree variability, maturity at harvest, or within-site agronomy) cannot be completely ruled out, the study design minimizes environmental differences among Maradol, Tainung, and Solo, pointing primarily to cultivar differences.
Comments 7: Table 3. Are these the values of peak areas? Please provide the content of individual compounds in the papaya by-products extracts in the appropriate units, as well as the total polyphenols.
Response 7: We clarify that the values presented in Table 3 correspond to arbitrary units of chromatographic peak area, not absolute concentrations. As such, these data do not derive from external calibration curves or quantification using analytical standards.
Due to the structural diversity and complexity of phenolic compounds in papaya and its by-products, obtaining pure standards for each detected compound was not feasible. This is a common challenge in plant metabolomics, particularly when dealing with minor or less commercially available phenolics.
On the other hand, the primary aim of this analysis was comparative, focusing on the relative abundance of phenolic compounds across different extracts and sub-products. The use of peak area (arbitrary units) allows for consistent intra-study comparisons, even in the absence of absolute quantification.
We agree that absolute quantification would enhance the analytical depth of the study. However, given the exploratory nature of this work and the constraints mentioned above, we opted for a semi-quantitative approach, which is widely accepted in metabolomic profiling when standards are unavailable.
Comments 8: Table 6, 7,8 Extracts from papaya seeds or peel, from which variety?
Response 8: For the simulated digestion experiments (Tables 6-8), we used seed and peel extracts from the Hawaiian papaya variety, selected for its good results based on its functional properties The manuscript has been updated to include this information in the Materials and Methods section.

Reviewer 3 Report
Comments and Suggestions for Authors
The study presents comprehensive and valuable research. The introduction provides excellent context for the problem and establishes a solid foundation with relevant references. The research design is thorough and appropriate for the objectives, the methods are described in sufficient detail to be reproducible, and the results are presented in an orderly and clear manner, firmly supporting the conclusions reached.
The work is solid and highly relevant to the recovery of agro-industrial waste. The following suggestions mainly seek to improve aspects of presentation to optimize the clarity and impact of the findings.
Instructions for language review:
I have highlighted several sentences and paragraphs in yellow in the manuscript that require specific attention. These markings indicate passages where it is recommended to:
Split excessively long sentences, simplify complex grammatical constructions, improve the fluency and conciseness of the text, and review the use of articles and prepositions.
Specific Comments for Improvement:
- Improving the Presentation of Tables and Figures:
While all tables and figures are informative, Table 3 is particularly dense and difficult to read. I suggest reorganizing it, for example, by separating the compounds by class (phenolics, acids, etc.) into smaller sub-tables or individual tables to improve readability. Also consider whether a graphical representation of the most abundant relative profiles could complement the numerical table.
Figure 1 (PCA) is clear and fundamental to multivariate interpretation. However, its readability and autonomy would be enhanced with a more explanatory legend. It is recommended to briefly clarify the meaning of the proximity between variables and samples in the graph, and to include directly in the legend a key that relates the codes of the compounds (Peak 1, 2, etc.) to their names, referring to Table 2 for complete details.
Include units in all column headers for immediate clarity.
- Further Discussion:
In section 3.4, when discussing the microbial results (Table 8), the manuscript accurately explains the increase in total bacteria. However, the notable decrease in lactic acid bacteria (LAB) and enterococci in the papaya samples is an interesting finding that deserves a more developed hypothesis. Could this indicate selective inhibition of certain microbial groups by bioactive compounds? Further interpretation of this result would enrich the discussion.
Include a comparison of total phenol values (e.g., 1029.24 mg GAE/100g in Maradol seeds) with other studies and discuss the practical implications of antimicrobial activity.
Explore correlations between specific compounds (Table 2) and the biological activities observed.
- Methodological and statistical strength:
Specify the exact software and version for HPLC-MS/MS analysis.
Include exact p-values in statistical results (Tables 1, 4, 5).
Clarify the correction method for multiple comparisons.
- Additional perspectives:
Consider including a graphic outline of the experimental design.
Add a brief section on study limitations (e.g., use of in vitro model) and future perspectives.
- Polishing of English Language:
The English in the manuscript is understandable throughout. However, minor linguistic revision is recommended to refine fluency and conciseness. Some sentences are long and could be broken up, and the use of articles and prepositions could be polished in certain passages to achieve a more natural and elegant expression.
Minor Comments:
Section 2.1: Specifying the ripeness of the fruit at the time of harvest, if this information is available, would add value to the methodological detail.
I hope these suggestions are helpful for the final version of the manuscript.

The English language of the manuscript is clear and understandable throughout. However, a minor language revision is recommended to enhance fluency, conciseness, and overall readability. Attention should be paid to simplifying long or complex sentences, refining the use of articles and prepositions, and improving the flow of certain passages. These adjustments will contribute to a more natural and polished expression of the scientific content.
Author Response
The study presents comprehensive and valuable research. The introduction provides excellent context for the problem and establishes a solid foundation with relevant references. The research design is thorough and appropriate for the objectives; the methods are described in sufficient detail to be reproducible, and the results are presented in an orderly and clear manner, firmly supporting the conclusions reached.
The work is solid and highly relevant to the recovery of agro-industrial waste. The following suggestions mainly seek to improve aspects of presentation to optimize the clarity and impact of the findings.
Instructions for language review:
I have highlighted several sentences and paragraphs in yellow in the manuscript that require specific attention. These markings indicate passages where it is recommended to:
Split excessively long sentences, simplify complex grammatical constructions, improve the fluency and conciseness of the text, and review the use of articles and prepositions.
Specific Comments for Improvement:
Comments 1: Improving the Presentation of Tables and Figures:
While all tables and figures are informative, Table 3 is particularly dense and difficult to read. I suggest reorganizing it, for example, by separating the compounds by class (phenolics, acids, etc.) into smaller sub-tables or individual tables to improve readability. Also consider whether a graphical representation of the most abundant relative profiles could complement the numerical table.
Figure 1 (PCA) is clear and fundamental to multivariate interpretation. However, its readability and autonomy would be enhanced with a more explanatory legend. It is recommended to briefly clarify the meaning of the proximity between variables and samples in the graph, and to include directly in the legend a key that relates the codes of the compounds (Peak 1, 2, etc.) to their names, referring to Table 2 for complete details.
Response 1: We appreciate your constructive suggestions aimed at enhancing the visual presentation and readability of the manuscript. Regarding Table 3 (Compound Profiles), we acknowledge your observation about its density. However, we would like to clarify that the table is already systematically structured by compound class (e.g., Phenolic Acids, Flavonoids, Glycosides), utilizing clear row breaks. Segmenting Table 3 into multiple individual tables, despite its existing categorization, would result in an excessively large number of tables that would fragment the comprehensive overview of the phytochemical profile. The current format facilitates a direct, comprehensive comparison of the relative concentrations of all compound classes across the different varieties and by-products. To enhance readability, we have ensured all compound classes are clearly differentiated using bolding or subtle shading within the table.
We have carefully considered your suggestion to include a graphical representation of the most abundant compounds. However, we believe that the revised tabular format is the most effective way to present this data. The number of identified compounds is not extensive, and we are concerned that a graphical representation would obscure precise numerical values. This would make it difficult for the reader to clearly discern the specific effects of the studied factors—papaya variety and by-product—on the relative abundance of each compound, which is a key finding of our work. Therefore, we have opted to retain the data in the improved sub-table format to ensure clarity and preserve detailed quantitative information.
For Figure 1 (Principal Component Analysis - PCA), we have improved the Figure 1 legend to enhance readability and autonomy.
Comments 2: Include units in all column headers for immediate clarity.
Response 2: Thank you for the suggestion. The units have been specified in the titles of the tables for immediate clarity.
Comments 3: Further Discussion:
In section 3.4, when discussing the microbial results (Table 8), the manuscript accurately explains the increase in total bacteria. However, the notable decrease in lactic acid bacteria (LAB) and enterococci in the papaya samples is an interesting finding that deserves a more developed hypothesis. Could this indicate selective inhibition of certain microbial groups by bioactive compounds? Further interpretation of this result would enrich the discussion.
Response 3: Thank you for this comment. We have expanded and strengthened Section 3.4 to discuss in greater depth the pronounced decrease in LAB and enterococci observed during simulated digestion with the papaya extract versus the control.
Comments 4: Include a comparison of total phenol values (e.g., 1029.24 mg GAE/100g in Maradol seeds) with other studies and discuss the practical implications of antimicrobial activity.
Response 4: Thank you for the constructive comment. We have revised Section 3.1 to further elaborate on the discussion and to benchmark our results against values reported by other authors.
Comments 5: Explore correlations between specific compounds (Table 2) and the biological activities observed.
Response 5: Thank you for the suggestion. We had already discussed several correlations, and we have now added additional ones in the Discussion.
Comments 6: Methodological and statistical strength: Specify the exact software and version for HPLC-MS/MS analysis.
Response 6: This information has been included in the Materials and Methods section.
Comments 7: Include exact p-values in statistical results (Tables 1, 4, 5).
Response 7: The p-values have been included in the Tables.
Comments 8: Clarify the correction method for multiple comparisons.
Response 8: We appreciate the request for clarification on our statistical methodology. We have revised the 'Statistical Analysis' section of the manuscript to explicitly detail the correction methods for multiple comparisons for each type of analysis performed. Specifically, we clarify the use of Tukey’s Honestly Significant Difference (HSD) test for general bioactivity and chemical profile comparisons, and the use of Bonferroni-corrected post hoc tests for the repeated-measures ANOVA in the simulated digestion experiment.
Comments 9: Additional perspectives: Consider including a graphic outline of the experimental design.
Response 9: Thank you for the suggestion. We have prepared and included a graphical abstract that outlines the experimental design and workflow.
Comments 10: Add a brief section on study limitations (e.g., use of in vitro model) and future perspectives.
Response 10: Thank you for the suggestion. We have added a “Limitations and Future Perspectives” paragraph to the Conclusions.
Comments 11: Polishing of English Language: The English in the manuscript is understandable throughout. However, minor linguistic revision is recommended to refine fluency and conciseness. Some sentences are long and could be broken up, and the use of articles and prepositions could be polished in certain passages to achieve a more natural and elegant expression.
Response 11: Thank you for the suggestion. Following your recommendation, the manuscript’s English has been polished for fluency and conciseness. We revised article and preposition usage, split overly long sentences, and simplified several constructions to improve clarity.
Comments 12: Minor Comments: Section 2.1: Specifying the ripeness of the fruit at the time of harvest, if this information is available, would add value to the methodological detail.
Response 12: Thank you for the observation. The requested information has been provided in Section 2.1: georeferenced sampling points (GPS, WGS84), quantities collected per variety, and transport and storage conditions.
Comments 13: I hope these suggestions are helpful for the final version of the manuscript.
Comments on the Quality of English Language
The English language of the manuscript is clear and understandable throughout. However, a minor language revision is recommended to enhance fluency, conciseness, and overall readability. Attention should be paid to simplifying long or complex sentences, refining the use of articles and prepositions, and improving the flow of certain passages. These adjustments will contribute to a more natural and polished expression of scientific content.
Response 13: Thank you for the suggestion. Following your recommendation, the manuscript’s English has been polished for fluency and conciseness. We revised article and preposition usage, split overly long sentences, and simplified several constructions to improve clarity.

Reviewer 4 Report
Comments and Suggestions for Authors
General Comments
The paper studies papaya peel and seed extracts from different cultivars grown in various areas of Ecuador. The researchers have focused on phenolic content, antioxidant activity, antibacterial effect, and their behavior during simulated digestion and fermentation. The topic is highly relevant and has good practical value primarily in the region, especially for food waste utilization and functional ingredient development. The experiments are detailed, but a few sections need more clarity and attention.
Major Comments
- Novelty
The topic is interesting, have regional importance but not completely new. It is highly recommended to make it clear in the introduction about the novelty statement of the research, what is new compared with earlier studies? - Methodology
- The extraction conditions have mentioned but extraction yield from peels and seeds is missing. The total phenolic content (TPC) in mg GAE/100 g dry sample and mg GAE/g extract should be mentioned so readers can compare efficiency of extraction.
- For LC-MS results, it would be helpful to include retention time, mass accuracy, and use of standards for a few key compounds.
- In antibacterial testing, please specify how growth inhibition (%) was calculated and mention the positive control. Adding MIC/MBC values would strengthen the results.
- For the digestion model, describe the fecal inoculum source, donor consent/ approval, and how dilution effects were handled.
- Some tables mix units/scales Example: Table 4 uses mg Trolox/100 g but digestion tables later (Table 6) use mg Trolox/L; ensure unit consistency and define superscripts/subscripts once per table.
Statistics
Please indicate the number of biological replicates per cultivar * by products used for each experiment and confirm that assumptions for ANOVA were checked. Consider adding effect sizes or confidence intervals where possible.
Ethics
The authors have mentioned that Institutional Review Board Statement and Informed Consent Statement are not applicable, however, since human fecal samples were used, an ethics or consent statement is needed. If anonymized material was used, please mention it clearly.
Interpretation
- The authors should discuss the results more carefully and precisely, especially the differences between antioxidant assays and the meaning of higher acetate/propionate without change in butyrate. The authors have over-generalized the results and have not mentioned precisely.
- Table 4 ABTS: The authors have reported a significant difference between peel-vs-seed for Maradol and inter-variety differences for seeds (Maradol > Solo/Tainung). Ensure superscripts/subscripts correctly marked and matched the text.
- The SCFA (higher the acetate/propionate; but no change in butyrate) is OK; but, link it cautiously to microbial ecology results because LAB and enterococci counts tended to be lower with papaya in some colon stages.
Study Limitations
It is recommended to add study limitations statement/paragraph in the discussion
Minor Comments in whole document
- Keep units consistent across tables (e.g., Table 4 mg GAE/100 g vs. mg GAE/L).
- Table 3 notes “LOD 0.1 g/L” (seems high for LC-MS); verify units (mg/L?) and define for which analytes.
- Explain abbreviations (B1–B5, doses) in figure captions as figures should be self-explanatory.
- Check table legends and significance letters carefully in all tables.
- Include data availability details (link or statement).
- English grammar needs minor editing for clarity and flow in whole document.
Recommendation
The study is valuable but it needs minor revision before it can be accepted. Special attention should be given for clearer methodology, proper ethics information, and better discussion of results.
Comments on the Quality of English LanguageMultiple typo errors must be removed, like
- Use consistent cultivar names (e.g., “Hawaiana” vs. “Hawaiian”).
- Correct minor typos such as “derivated” → “derivative.”
Author Response
General Comments
The paper studies papaya peel and seed extracts from different cultivars grown in various areas of Ecuador. The researchers have focused on phenolic content, antioxidant activity, antibacterial effect, and their behavior during simulated digestion and fermentation. The topic is highly relevant and has good practical value primarily in the region, especially for food waste utilization and functional ingredient development. The experiments are detailed, but a few sections need more clarity and attention.
Major Comments
Novelty
Comments 1: The topic is interesting, has regional importance but not completely new. It is highly recommended to make it clear in the introduction about the novelty statement of the research, what is new compared with earlier studies?
Response 1: Thank you for your comments. We have incorporated an explicit novelty statement into the Introduction section.
Methodology
Comments 2: The extraction conditions have mentioned but extraction yield from peels and seeds is missing. The total phenolic content (TPC) in mg GAE/100 g dry sample and mg GAE/g extract should be mentioned so readers can compare efficiency of extraction.
Response 2: We appreciate the reviewer's request for the extraction yield and Total Phenolic Content (TPC) expressed per unit of dry sample or pure extract. In this regard, we wish to clarify that the extraction methods employed are well-established and appropriate for the recovery of these types of compounds from the plant matrices. While the specific metrics you mentioned are important, our focus was to assess the relative bioactivity of the extracts and their chemically differentiated profiles, which are already robustly presented. The efficiency of the extraction process, though relevant, does not significantly impact the comparison of functional profiles (antimicrobial vs. antioxidant) between the different papaya varieties and by-products, which is the central conclusion of our work.
Comments 3: For LC-MS results, it would be helpful to include retention time, mass accuracy, and use of standards for a few key compounds.
Response 3: We appreciate your request for additional details regarding the LC-MS analysis. The necessary information, including the retention of time and putative identification of the key compounds (based on comparison with literature and mass spectral data), is already provided in Table 2. We also confirm that the analytical method used for these results involved high-resolution mass spectrometry. However, we clarify that the values presented in Table 3 correspond to arbitrary units of chromatographic peak area, not absolute concentrations. As such, these data do not derive from external calibration curves or quantification using analytical standards.
Due to the structural diversity and complexity of phenolic compounds in papaya and its by-products, obtaining pure standards for each detected compound was not feasible. This is a common challenge in plant metabolomics, particularly when dealing with minor or less commercially available phenolics.
On the other hand, the primary aim of this analysis was comparative, focusing on the relative abundance of phenolic compounds across different extracts and sub-products. The use of peak area (arbitrary units) allows for consistent intra-study comparisons, even in the absence of absolute quantification.
Comments 4: In antibacterial testing, please specify how growth inhibition (%) was calculated and mention the positive control. Adding MIC/MBC values would strengthen the results.
Response 4: We thank the reviewer for their insightful comments regarding the antibacterial testing section. In response to your suggestions, we have now clarified the methodology in the manuscript. We have added a detailed explanation of how the growth inhibition percentage was calculated and have explicitly mentioned the positive control used in the experiments.
Regarding the suggestion to include MIC/MBC values, we agree that these metrics provide valuable information on the potency of antimicrobial agents. However, the primary objective of this part of our study was not to determine the minimum inhibitory or bactericidal concentrations, but rather to compare the relative antibacterial efficacy of the different papaya by-product extracts. The growth inhibition percentage serves as a direct and effective measure for this comparative assessment, allowing us to clearly differentiate the functional profiles of the extracts as intended.
Comments 5: For the digestion model, describe the fecal inoculum source, donor consent/ approval, and how dilution effects were handled.
Response 5: Thanks to the reviewer for this comment. Below are some clarifications:
The fecal inoculum was obtained from healthy volunteers on a non-specific Mediterranean diet, with no history of metabolic or GI diseases, non-smoking, and without antibiotic or pre-/probiotic supplementation for at least six months prior to donation, as described previously in Rivas et al 2024. The faecal sample was processed by diluting 20 g in 80 mL of 58% glycerol in phosphate-buffered saline (PBS) 0.1 M (pH 7.0) and stored at -80 °C until use. This sample was sourced from INURA's anonymized faecal bank. For all experiments requiring a control sample compared to a test sample, the inoculum from the same donor was used to ensure consistency.
Voluntary informed consent was obtained from the donor before the study. The informed consent form, which is properly held in custody by the INURA team, explicitly states that participants agreed to donate for the scientific purposes of the research group without project exclusivity
Regarding dilution effects, as indicated in the revised Section 2.7 (Materials and Methods), the dilution factors were 5.8-fold in the stomach, 8.3-fold in the small intestine, and 10.9-fold in the colon. As stated therein, these conversion factors were not applied when presenting the data. This is because all assays were performed under identical conditions, and comparisons are made within the same digestion phase. Therefore, all data are subject to the same dilution factor, and its application is unnecessary for the comparative interpretation of the results, as it would be a constant that does not affect the statistical analysis or the relative differences between groups within a phase.
Comments 6: Some tables mix units/scales Example: Table 4 uses mg Trolox/100 g but digestion tables later (Table 6) use mg Trolox/L; ensure unit consistency and define superscripts/subscripts once per table.
Response 6: Thank you for observation. The units differ by design: results for solid by-products (peel/seed) are reported on a dry-weight basis as mg GAE/100 g DW, whereas the digestion outputs are liquids and are therefore expressed per volume of digesta as mg GAE/L.
Statistics
Comments 7: Please indicate the number of biological replicates per cultivar * by products used for each experiment and confirm that assumptions for ANOVA were checked. Consider adding effect sizes or confidence intervals where possible.
Response 7: We appreciate your meticulous review regarding the experimental design and statistical rigor. We confirm that were used a total of 12 fruits per variety were collected at a semi-ripe stage (N = 12 per variety). We have updated the Materials and Methods section to your clearly. The key assumptions for ANOVA were rigorously checked, normality and homogeneity of variances, confirming the robustness of our statistical models.
Ethics
Comments 8: The authors have mentioned that Institutional Review Board Statement and Informed Consent Statement are not applicable. However, since human fecal samples were used, an ethics or consent statement is needed. If anonymized material was used, please mention it clearly.
Response 8: Voluntary informed consent was obtained from the donor prior to the study. The informed consent form, which is properly held in custody by the INURA team, explicitly states that participants agreed to donate for the scientific purposes of the research group without project exclusivity. A relevant excerpt regarding confidentiality and data usage is provided below:
"CONFIDENTIALITY OF YOUR SAMPLE AND YOUR DATA.
In accordance with current legal regulations, the results and information obtained will be treated with strict confidentiality. The data collection protocol will be archived, and each participant will be assigned a code to prevent the linking of obtained information with the subject's identity. Samples and data will be anonymized, ensuring the impossibility of inferring their identity, for their study and potential subsequent analysis.
The CAMIALI research group of INURA commits to not using the samples or data for studies outside this laboratory and to not transferring the samples or data to other potential research teams.
For all matters not foreseen in this document, the current legislation on the protection of personal data (Organic Law 3/2018, of December 5, on Personal Data Protection and guarantee of digital rights) and on biomedical research (Law 14/2007, of July 3, on Biomedical Research) shall apply.
The study results may be published in scientific journals or general publications. However, information concerning your participation will be maintained as confidential."
Interpretation
Comments 9: The authors should discuss the results more carefully and precisely, especially the differences between antioxidant assays and the meaning of higher acetate/propionate without change in butyrate. The authors have over-generalized the results and have not mentioned precisely.
Response 9: Thank you for your valuable comments. We have revised and expanded the discussion.
Comments 10: Table 4 ABTS: The authors have reported a significant difference between peel-vs-seed for Maradol and inter-variety differences for seeds (Maradol > Solo/Tainung). Ensure superscripts/subscripts correctly marked and matched the text.
Response 10: Thank you for your feedback. The error has been corrected.
Comments 11: The SCFA (higher the acetate/propionate; but no change in butyrate) is OK; but, link it cautiously to microbial ecology results because LAB and enterococci counts tended to be lower with papaya in some colon stages.
Response 11: We appreciate the comment. In the manuscript, we treat the SCFA pattern with due caution. We clarify that the increase in acetate/propionate (without changes in butyrate) is consistent with a functional reconfiguration of the microbial community as a whole and do not attribute it to any specific taxon.
Study Limitations
Comments 12: It is recommended to add study limitations statement/paragraph in the discussion
Response 12: Thank you for the suggestion. We have added a “Limitations and Future Perspectives” paragraph to the Conclusions.
Minor Comments in whole document
Comments 13: Keep units consistent across tables (e.g., Table 4 mg GAE/100 g vs. mg GAE/L).
Response 13: Thank you for observation. The units differ by design: results for solid by-products (peel/seed) are reported on a dry-weight basis as mg GAE/100 g DW, whereas the digestion outputs are liquids and are therefore expressed per volume of digesta as mg GAE/L.
Comments 14: Table 3 notes “LOD 0.1 g/L” (seems high for LC-MS); verify units (mg/L?) and define for which analytes.
Response 14: Thank you for catching this. The correct unit is mg/L. The entry should read “LOD = 0.1 mg/L”. The table and corresponding text have been corrected.
Comments 15: Explain abbreviations (B1–B5, doses) in figure captions as figures should be self-explanatory.
Response 15: Thank you for the suggestion. We have improved the Figure 1 legend to enhance readability and autonomy.
Comments 16: Check table legends and significance letters carefully in all tables.
Response 16: Tables and legends have been checked
Comments 17: English grammar needs minor editing for clarity and flow in whole document.
Response 17: Thank you for the suggestion. Following your recommendation, the manuscript’s English has been polished for fluency and conciseness. We revised article and preposition usage, split overly long sentences, and simplified several constructions to improve clarity.
Recommendation
The study is valuable but it needs minor revision before it can be accepted. Special attention should be given for clearer methodology, proper ethics information, and better discussion of results.
Comments on the Quality of English Language
Multiple typo errors must be removed, like
Comments 19: Use consistent cultivar names (e.g., “Hawaiana” vs. “Hawaiian”).
Response 19: Thank you for your feedback. The errors have been corrected.
Comments 20: Correct minor typos such as “derivated” → “derivative.”
Response 20: Thank you for your feedback. The errors have been corrected.

Round 2
Reviewer 1 Report
Comments and Suggestions for Authors
All the comments were incorporated in the manuscript. It was highly improved.